# The Contribution of *JAK2* 46/1 Haplotype in the Predisposition to Myeloproliferative Neoplasms

**DOI:** 10.3390/ijms232012582

**Published:** 2022-10-20

**Authors:** Jhemerson Paes, George A. V. Silva, Andréa M. Tarragô, Lucivana P. de Souza Mourão

**Affiliations:** 1Programa de Pós-Graduação em Ciências Aplicadas à Hematologia, Universidade do Estado do Amazonas (UEA), Manaus 69850-000, AM, Brazil; 2Fundação Hospitalar de Hematologia e Hemoterapia do Amazonas (FHEMOAM), Manaus 69050-001, AM, Brazil; 3Fundação Oswaldo Cruz–Instituto Leônidas e Maria Deane (Fiocruz), Manaus 69027-070, AM, Brazil

**Keywords:** JAK2 germline haplotype, myeloid neoplasms, haplotype, molecular pathogenesis, single nucleotide polymorphisms

## Abstract

Haplotype 46/1 (GGCC) consists of a set of genetic variations distributed along chromosome 9p.24.1, which extend from the Janus Kinase 2 gene to Insulin like *4*. Marked by four jointly inherited variants (rs3780367, rs10974944, rs12343867, and rs1159782), this haplotype has a strong association with the development of *BCR-ABL1*-negative myeloproliferative neoplasms (MPNs) because it precedes the acquisition of the JAK2V617F variant, a common genetic alteration in individuals with these hematological malignancies. It is also described as one of the factors that increases the risk of familial MPNs by more than five times, 46/1 is associated with events related to inflammatory dysregulation, splenomegaly, splanchnic vein thrombosis, Budd–Chiari syndrome, increases in RBC count, platelets, leukocytes, hematocrit, and hemoglobin, which are characteristic of MPNs, as well as other findings that are still being elucidated and which are of great interest for the etiopathological understanding of these hematological neoplasms. Considering these factors, the present review aims to describe the main findings and discussions involving the 46/1 haplotype, and highlights the molecular and immunological aspects and their relevance as a tool for clinical practice and investigation of familial cases.

## 1. Introduction

Myeloproliferative neoplasms (MPNs) consist of a set of hematological cancers that are characterized by hyperplasia of one or more elements of the myeloid series (leukocytes, platelets, and red blood cells) with effective maturation, proliferation [1,2,3] and the possibility of progression to medullary fibrosis or leukemic transformation [4]. The global incidence is six cases per 100,000 individuals [5], affecting mostly individuals between 60 and 70 years old, and is more prevalent in white males [3,5].

For MPNs, the WHO Classification of Tumors of Hematopoietic and Lymphoid Tissues—5th edition, 2022, classifies the following hematological malignancies: chronic myeloid leukemia (CML), polycythemia vera (PV), essential thrombocythemia (ET), primary myelofibrosis (PMF), chronic neutrophilic leukemia (CNL), chronic eosinophilic leukemia (CEL), juvenile myelomonocytic leukemia (JMML), and myeloproliferative neoplasm, not otherwise specified (MPN-NOS) [6,7]. The MPN *BCR-ABL1*—negative [5]. The PV, ET, and PMF are the most frequent (Table 1), and share genetic variations that constitutively activate the physiological signal transduction pathways responsible for hematopoiesis, which leads to an increase in myeloid proliferation, though without impairing maturation and cell differentiation [3,5,8,9].

In MPNs, the JAK-STAT pathway plays an important role in the signaling of cytokines and growth factors, which act in the regulation of cell proliferation, differentiation, survival, immune response, and oncogenesis [22,23,24]. Previous studies have linked prolonged activation of JAK-STAT signaling with aberrant hematopoietic stem cell development and hematologic malignancies [25,26,27,28]. These alterations are associated with the presence of gain-of-function genetic variants in the *JAK2* gene, which encodes the protein of the same name. These variants cause constitutive activation of the pathway, resulting in myeloproliferation and cytokine production, which is the definitive phenotype of MPNs [26,27,29]. 

## 2. *Janus Kinase* Gene (*JAK2*)

The *Janus kinase 2* gene (HGNC ID: 6192) is located on chromosome 9p24.1 [30], and has 142,939 base pairs (bp) in which the promoter region, 25 exons, 25 introns, and the terminator region are located. The coding DNA sequence (CDS) is composed of 3399 nucleotides distributed between exon 3 and 25, as established by its reference sequence (RefSeq.: NG_009904.1; NM_001322194.2), which is made available by the National Center for Biotechnology Information (NCBI). This gene presents alternative splicing, giving rise to seven transcripts of sizes that vary from 6900 to 7000 bp, and encodes three isoforms (A, B and C) of the Janus kinase 2 (JAK2) protein [30,31,32].

The JAK proteins consist of a family of nonreceptor cytoplasmic kinases that encompass four mammalian protein types: JAK2, which is part of the signaling of homodimeric receptors, such as the erythropoietin receptor (EPOR), the thrombopoietin receptor (MPL), and the granulocyte colony stimulating factor (G-CSFR), which are also used by some heterodimeric receptors; and JAK1, JAK3, and tyrosine kinase 2 (TYK2), which are useful in signaling heterodimeric receptors [21,27,29]. These proteins are relatively large and have approximately 1150 amino acids and a molecular weight ranging from 116 to 140 kDa [33]. JAK2 deserves attention due to its role in the hematopoietic proliferation mechanism, especially in relation to MPNs.

JAK2 consists of four domains: two kinase domains, JH1 (tyrosine kinase, catalytically active, and located in the C-terminal portion) and JH2 (pseudokinase and catalytically inactive), which is responsible for inhibiting the JH1 domain and promoting cytokine-dependent activation; a FERM-like domain (4.1/ezrin/radixin/moesin) constituted by the homologous domains JH5, JH6, and JH7, located in the N-terminal portion, which is responsible for the noncovalent binding of JAKs to the cytokine receptor; and an SH2-like domain (Src Homology 2), which contains the JH4 and JH3 homologous domains [21,34]. Experimental studies reveal that the homozygous germline deletion of JAK2 alleles results in embryonic lethality due to ineffective erythropoiesis, since JAK2-deficient hematopoietic progenitors do not respond to erythropoietin stimulation [27,29]. This highlights the importance of genetic aspects involving the *JAK2* gene, its locus, and the events that affect this region, such as acquired uniparental disomy of chromosome 9p.

### 2.1. Acquired Uniparental Dysomy

Oncological diseases are characterized by instability and the gradual accumulation of genetic alterations over time [35], which are caused by genetic events intrinsic to the cell or by exposure to external mutagens [36]. Uniparental disomy (UPD) is one of these alterations, and is recognized as a hallmark of cancer genomes [35].

UPD was described in 1980 [37] and is defined as the occurrence of the inheritance of two homologous chromosomes from the same parental origin [38], and is caused by segregation errors in meiosis I or meiosis II [35]. The latter gives rise to isodysomy [39], in which the affected region is genetically identical, thus, resulting in the development of several genetic disorders through the gain or loss of chromosomal regions, or by the presence of two identical copies of abnormal genes or nucleotide sequences [38]. This event also occurs in somatic cells, and receives the nomenclature of acquired uniparental disomy (aUPD). In it, adventitious genetic variants are amplified, and lead to a growth advantage through the conversion of a heterozygous cell into a homozygous cell, with no change in the number of DNA copies [35,40,41,42,43,44,45,46]. Two possible mechanisms can lead to the occurrence of aUPD: (1) nondisjunction of chromatids (cells with the same originally duplicated chromosome are generated); or loss of chromosomes due to delay in mitotic anaphase [39] and, in an attempt to balance the loss of a chromosomal molecule, a duplication of the remaining chromosome as a template is made, resulting in two identical chromosomes. Another possibility is (2) reciprocal exchange of chromosomal material during mitosis (mitotic recombination), such as chromatids, thus generating several possible results [35,39] (Figure 1).

Also known as copy number neutral loss of heterozygosity (CNN-LOH) [47,48], aUPD was first identified by Kralovics [41] in patients with PV, and describes a mitotic recombination associated with neutral loss of heterozygosity of chromosome 9p [24]. Over the years, this abnormality has been identified in several loci in a variety of neoplasms. Its impact is the conversion of genetic variations to the homozygous state in essential genes, such as *JAK2* and *CDKN2A* at 9p, *FLT3* in 13q, *TP53* in 17p, and others, including *WT1, CBL, RUNX1, *and *TET,* which are related to the initial process or progression of these diseases [43]. In the context of MPNs, more specifically in PV, this alteration proved to be a common finding, as in other hematological malignancies [35,39,43], and defines the molecular scenario of MPNs [49], with the JAK2V617F variant being reported as present in most patients with MPN [48,50].

### 2.2. JAK2V617F Variant

Discovered in 2005 by Kralovics [48], the JAK2V617F variant (dbSNP ID: rs77375493) revolutionized the genetic knowledge and diagnosis of MPNs [51]. JAK2V617F, an acquired somatic variant with gain of function with exchange of a guanine (G) for thymine (T) at nucleotide 1849 (c.1849G>T) of exon 14 of JAK2, results in the substitution between valine (V) and phenylalanine (F) at position 617 of the polypeptide chain, which affects the JH2 domain of the protein [27,31]. The variant affects JH2 auto-inhibitory activity, with constitutive activation of JH1 as a result and, consequently, of the JAK-STAT pathway, which interferes with intracellular signaling [14]. JAK2V617F causes the transformation of hematopoietic cells into cytokine-independent growth, thus, promoting tumorigenesis, tumor progression, and inflammation caused by continuous stimulation within the hematopoietic cell [17,26,52,53,54].

JAK2V617F is the most common genetic event in *BCR-ABL1*-negative myeloproliferative neoplasms. It is present in >95% of patients with PV and 50–60% of patients with PMF and ET [3,14,55,56], and is considered as a criterion for diagnosis by the WHO [5,7]. Patients with JAK2V617F negative PMF and ET may have other genetic alterations in exons 10 and 9 of the *MPL* and calreticulin (*CARL*) genes, respectively [57,58,59]. Leucine to lysine substitutions at codon 539, glutamic acid deletions at codon 543 and insertions leading to the substitution of phenylalanine at codon 547 have already been identified in exon 12 of the* JAK2* gene of patients with PV [51,60,61,62,63,64], which demonstrates the complexity of the genetic scenario involved in MPNs.

Different studies have highlighted the origin of JAK2V617F in a multipotent hematopoietic stem cell [29,50,65], which provides a selective advantage over the normal multipotent hematopoietic cell, and results in myeloid differentiation and a myeloproliferative phenotype [24,50]. As a result, the abnormal myeloid clone proliferates and interrupts the medullary microenvironment, which promotes a malignant niche that favors stem cells with genetic alterations in relation to normal ones, and leads to an eventual mobilization of mature cells to the peripheral blood. This explains the presence of the variant in leukocytes in genetic analysis [50,54,65,66]. Furthermore, activation of the JAK-STAT pathway is more evident in patients with a high load of the variant JAK2V617F allele, thus, demonstrating that there are differences in signaling based on the presence of heterozygous or homozygous JAK2V617F [24,55]. 

Approximately one third of JAK2V617F positive PV and PMF cases are homozygous with variant allele loads greater than 50%, whereas in ET it is lower (approximately 25%) and close to 100% in post-PV or post-ET patients [14]. In PV, patients that are homozygous have a longer disease duration and a risk of progression to myelofibrosis [67]. JAK2V617F homozygosity is a consequence of aUPD, which accompanies the variant and reduces it (and any allele that is in linkage disequilibrium) to a homozygous state, leading to duplication of the mutated allele and consequent loss of the unmutated allele [38,45,47,67,68,69,70]. This relationship between the two events (homozygosity and aUPD) raises the question of how distinct genetic mechanisms can gradually correlate in order to increase the allelic dosage of a known gain-of-function genetic variant in MPNs, with the potential to lead to oncogenic transformation of an aggressive premalignant clonal cancer, such as a leukemic transformation [45].

As described, JAK2V617F leads to clonal proliferation in MPNs; however, it is not clear which factors influence the development, severity and phenotype of the disease [66]. The latter is possibly related to individual characteristics (sex, associated inflammatory disease), and genetic abnormalities (driver genes, pathogenic genetic variants and other chromosomal aberrations) [71]. Different signaling pathways, epigenetic modulation, immune system, lifestyle, JAK2V617F variant allele load, and exceptional germline alleles found in population-wide and hereditary cases are other possible factors involved in the development of JAK2V617F73-associated MPNs [71,72]. The discovery of this genetic alteration has brought benefits for the therapy and diagnosis of MPNs; however, some questions remain unclear, such as the events that precede its acquisition, since it is not a germline genetic variant [29]. New speculations have arisen with the discovery of the 46/1 haplotype, whose studies aim to clarify most of these questions.

## 3. 46/1 Haplotype

Humans are diploid organisms with two copies of each chromosome, similar to each other and differ only in a small fraction of information (0.1%) [73]. These differences are contained in sites of single-base genetic alterations called single nucleotide variants (SNVs), which contribute to interindividual and inheritable differences in complex phenotypes [74,75]. A group of genetic variations present on the same chromosome, which are not easily separable by recombination and therefore tend to be inherited together, is called a haplotype [76].

Most of these variants make up a haplotype and are in linkage disequilibrium (LD), a nonrandom association of alleles at two or more loci that exists because of shared ancestry of contemporary chromosomes [77]. This is related to the timing of variant events and genetic distance, and can provide valuable information on the location of disease variants from genetic markers [77,78,79,80]. SNVs within a haplotype block originally arose from a single historical event of genetic variation and, therefore, are associated with closely related variants that were present on the ancestral chromosome in which these changes occurred [81]. For this reason, and other reasons, strong statistical associations between genetic variants are described, and the presence of a particular variant at one site can predict or “mark” the presence of a specific variant at another locus (carrying too much genetic information) [75].

Haplotypes have become useful tools in genetic investigation thanks to the efforts established by HapMap International [77,80,81,82] and the 1000 Genomes Project Consortium [73,83,84]. The data obtained from these initiatives can be used for studying human migration, evolutionary selection, population structure, imputation of intronic regions, and understanding of the genetic association between pathological variants [85].

Between 2008 and 2009, lines of investigation involving haplotypes and MPNs were reported by different research groups [86,87]. Haplotype 46/1 was the first set of germline risk variants described in MPN and one of the first signs of hereditary predisposition, also associated with cases of splenomegaly, splanchnic venous thrombosis (SVT), increased hematocrit and Budd–Chiari syndrome in patients positive for JAK2V617F [88,89,90,91], inflammatory bowel disease [92], ulcerative colitis [93] and Crohn’s disease in patients without MPNs [93,94,95].

The nomenclature of this haplotype was first described by Jones et al. [24], who observed 109 cases of identical haplotypes of the *JAK2* gene in 142 alleles when the JAK2V617F variant was present. As residual wild-type alleles, the haplotype was identified in only 12% of cases. These propositions demonstrated that the loss of JAK2V617F heterozygosity is not random, and happens in a specific* JAK2* haplotype. In order to expand the understanding of these data, Jones et al. [24] selected 14 SNVs, which resulted in 92 possible haplotypes. Of these, two (numbers 46 and 1, collectively referred to as 46/1) were identical and frequent in JAK2V617F positive patients compared to controls.

Consisting of hundreds of variants, this haplotype extends over a linkage disequilibrium block with a length between 250–280 Kb of chromosome 9p.24.1, which encompasses three genes: *JAK2*, *Insulin like 6* (*INSL6*—RefSeq.: NG_046969.1; HGNC ID: 6089) and *Insulin like 4* (*INSL4*—RefSeq.: NC_000009.12; HGNC ID: 6087); the latter two are not expressed in the hematopoietic system [47,66,71,88,96] (Figure 2).

Different SNVs were mapped, with the vast majority being identified in Table 2 and Figure 3. Some are used to identify the 46/1 haplotype. The genetic alterations work only as markers, and the true causal variants still remain poorly known or totally hidden in the LD [47] block. The following four SNVs in LD are considered the most studied markers of the haplotype: rs3780367 (NG_009904.1:g.83511T>G), rs10974944 (NG_009904.1:g.85587C>G), rs12343867 (NG_009904.1:g.88945T>C), and rs1159782 (NG_009904.1:g.92873T>C), which are located at introns 10, 12, 14, and 15, respectively. The minor allele frequency (MAF) [83] is shown in Figure 4. rs10974944 was the first to be associated with the emergence of MPNs [98]. Studies carried out in Europe, Japan, China, North America, and Brazil have shown that the variant allele of rs10974944 (G) is more frequent in all MPN patients (especially those positive for JAK2V617F) than in the control population [68,76,89,98,99,100].
ijms-23-12582-t002_Table 2Table 2Single nucleotide variants (SNVs) identified in studies on the 46/1 haplotype and their respective information described in the literature.SNVReferencesConclusions
rs10974944
[68,76,89,97,98,99,100,101,102]Studies carried out in populations of Brazilian, Japanese, and Chinese origin; this variant has a strong association with JAK2V617F positive MPN patients when compared to controls; rs10974944 (G) is a risk allele for MPNs.
rs12686652
[89]Significantly associated with patients with PV in this case-control study, but no association with MPNs in the Japanese population.
rs12335546

rs12343867
[71,89,90,99,100,101,102]Associated with positive JAK2V617F in the populations of Japan, China, and Taiwan, especially in individuals with PV; this is used as a haplotype marker. Association with splenomegaly has been reported and is in LD with other SNVs of haplotype 46/1.
rs4495487
[89]More frequent in PV patients in a case-control study in Japan. It has not been reported in Caucasian populations and may contribute to the MPN phenotype in the Japanese population.
rs691857
[101]No significant association.
rs17803986

rs7848509

rs10758677

rs3780367
[103,104]In linkage disequilibrium with other markers of the haplotype and has significant association with MPNs, but no population data.
rs12340895
[100]Associated with JAK2V617F positive MPNs in Chinese patients.
rs12342421
[100]Associated with the predisposition to develop JAK2V617F positive MPNs (OR = 3.55) in carriers for the minor C allele (in Chinese populations) with a 250% increased risk for disease, regardless of haplotype 46/1.
rs1159782
[99,104]It is in linkage disequilibrium with markers of the 46/1 haplotype.
rs10119004
[100]Associated with positive JAK2V617F and reported for the first time in the same study
rs12343065

rs10815162

rs7857730

rs7847294

rs3780378

rs2149556

rs2149555

rs1887428
[103]Able to alter the expression rate of *JAK2*.


The rs10119004 variant (NG_009904.1:g.:85805G>A; MAF of G: 38%), located close to rs10974944, was cited in studies involving MPNs, and was associated with MPNs in the Chinese population [100].

There is a wide academic-scientific discussion regarding the variants present in the haplotype, and these genetic alterations cannot be considered as the only cause of clonal proliferation, since, for the development of an MPN, there is a need to acquire additional somatic variants, such as the JAK2V617F variant [47]. Therefore, 46/1 is described as one of the possible “pre-JAK2V617F” events, which is a predisposition factor that is strongly linked to three to four times higher chances of development of MPNs and responsible for half of the risk of MPNs attributable to inherited factors [69,109,110,111].

## 4. Association between the 46/1 Haplotype and the JAK2V617F Variant

The acquisition of somatic variants is a pathogenic mechanism of great importance in the development of MPNs, and genetic antecedent factors also play an important role in their development [89]. In the context of these hematological malignancies, a possible association between the 46/1 haplotype and the JAK2V617F variant has been described, for example, in the study carried out by Kilpivaara [98], which identified (1) the rs10974944 variant (C/G) in the *JAK2* gene, which predisposes the development of JAK2V617F positive MPNs, (2) three MPN modifier loci unknown at the time of the study, and reported that (3) JAK2V617F acquisition is preferentially acquired in cis with the predisposing allele, and that (4) rs10974944 and JAK2V617F are located in a common haplotype block that does not span the *JAK2* 5′ promoter (they are not in LD [101]), thus, the rs10974944 (G) allele may predispose the JAK2V617F somatic variant on the same strand [98].

Likewise, the germline findings identified support the hypothesis that 46/1 contributes to the predisposition of MPNs. These findings are in agreement with the reports by Olcaydu [101], in which the haplotype rs3780367G/rs10974944G/rs12343867C/rs1159782C was strongly associated with JAK2V617F [101].

### Haplotype 46/1 Agreement with JAK2V617F in Different Populations

Previous studies have reported the association of 46/1 haplotype variants with JAK2V617F in ethnically distinct populations. In one study performed in China, a significant association was described between the JAK2V617F variant and the rs10974944 (G) of haplotype 46/1, with a higher frequency being observed in patients after comparing them with controls [68]. Similar results were observed in Japan by Ohyashiki [89], who evaluated 138 patients and 107 healthy subjects aged 30–87 years, and highlighted the JAK2V617F status in the patients (68.8% JAK2V617F positive) and 107 control subjects. Thus, the combination allele G at rs10974944, allele C at rs4495487, and allele C at rs12343867 was strongly associated with MPN positive JAK2V617F (OR: 3.07; 95% CI: 1.73–5.46) and discretely associated with MPN JAK2V617F negative (OR: 2.26, 95% CI: 1.01–4.7) when compared to controls. This demonstrates that carriers of 46/1 have a 200–300% increased risk (2–3 times more likely) of acquiring JAK2V617F when compared to noncarriers. These findings are in agreement with those of Tefferi [69], Triffa [112], Jones [24], Kilpivaara [98], Olcaydu [101], Pardanani [87], and Wang [45] who carried out studies with Caucasian populations from the United States of America and several European countries.

Another study carried out in China evaluated an SNV in LD (rs12340895) with haplotype 46/1 in 225 patients and 226 controls, as it was identified as a risk factor for MPNs, as well as homozygosity at the rs12340895 locus as a factor of susceptibility to JAK2V617F [113]. Similar results were found in the population of Taiwan with SNV marker rs1234387 [90]. The different reports in different populations around the world highlight that the mechanism underlying the acquisition of JAK2V617F is not limited to Caucasians only; therefore, it must have a relatively ancient evolutionary basis [47].

There are two hypotheses that could explain the association between the 46/1 haplotype and the JAK2V617F variant: hypermutability and fertile soil hypothesis [24,45,114]. The hypermutability hypothesis considers 46/1 as more genetically unstable [68], with the possibility of leading to DNA damage and replication errors [88] as it predisposes one to the acquisition of JAK2V617F more frequently when compared to other haplotypes [96]. Support for this hypothesis comes from the observation that JAK2V617F apparently appeared at least twice in some individuals, and possibly because exon 12 variants are associated with 46/1, albeit at a lower risk [47,115]. On the other hand, the fertile soil hypothesis assumes that hematopoietic stem cells carrying 46/1 have a selective advantage when oncogenic variants occur [96,99]. Even with different propositions, one hypothesis does not cancel out the other and both can coexist in the genetic scenario of these neoplasms [76].

## 5. Contribution of 46/1 to Inflammatory Dysregulation in MPNs

Hematopoietic stem cells require a set of tightly regulated and conserved cooperative interactions with their stromal cells in order to carry out the normal processes of dormancy, self-renewal, proliferation, locomotion, and differentiation. These depend on the expression of hematopoietic genes, interaction between cells, production, and release of a variety of cytokines and chemokines [116] related to the inflammatory mechanisms involved in MPNs. In our previous review, we identified that the JAK2V617F variant plays a relevant role in this complex process by interfering with the regulation of several pathways involved in the production of cytokines, tumorigenesis, and inflammation mediators [56].

Studies suggest that the haplotype is related to an elevated expression of *JAK2, INSL6,* and *INSL4*, which causes DNA recombination, emergence of genetic variations, or abnormal methylation of the promoter region [97,117]. Hermouet [97] suggests that 46/1 may include unidentified intronic repeating DNA sequences that facilitate DNA recombination and overexpression of the *JAK2 *gene located on the recombined allele. In this context, JAK2 transmits the proliferation signals of all cytokines critical to myelopoiesis, and the 46/1 haplotype would predispose carriers to chronically excessive stimulation of myelopoiesis. This exposes myeloid progenitors to an exacerbated mitotic process, and increases the risk of error and alteration in myelopoiesis-directed genes, such as* JAK2 *and* MPL*, *TET2*, *ASXL1*, *LNK*, *CBL*, and *EZH2* [97]. One study showed that MPN positive patients present a high expression of mRNA of *JAK2*, which would be related to a greater probability of myeloid cells dividing in response to the protein activating cytokines, thus, making them prone to replication errors [97]. In addition, the haplotype may, in theory, contribute to a preponderant downstream signaling of constitutively activated JAK2V617F through increased cytokine production by bone marrow stromal cells, possibly mediated by *INSL4* and *INSL6* [71,97,117]. The latter has already been reported to be expressed in rat medullary stromal cells [118].

The haplotype can also influence the acquisition of somatic variants in *JAK2*, as well as facilitate the expression of *INSL6* and *INSL4 *in medullary stromal cells, which leads to abnormal signaling of cytokines with proinflammatory and promyeloid action, and generates a favorable environment for the mutated clone (Figure 5). It is not known for sure which cytokines would be related; however, several clinical studies with MPN patients have already demonstrated increased plasma levels of IL-1, IL-2, IL-6, IL-8, IL-12, TNF-α, and IFN-γ and growth factors, including granulocyte-macrophage colony-stimulating factor (GM-CSF), platelet-derived growth factor (PDGF), and vascular endothelial growth factor (VEGF) [119,120]. In PMF, cellular and extracellular levels of several cytokines with angiogenic and fibrinogenic action, such as transforming growth factor beta (TGF-ß), platelet-derived growth factor (PDGF), basic fibroblast growth factor (bFGF), and vascular endothelial growth factor (VEGF), are increased, among others that condition the medullary stroma to create a favorable pathological microenvironment that nourishes and protects malignant cells via histological alterations of bone marrow [121,122]*.* It can be hypothesized that the haplotype acts as a possible factor in the genetic susceptibility of the host to an inadequate myeloid response to cytokines, thus leading to an intensified inflammatory state and increased risk of myeloid neoplasms, which is accelerated by the acquisition of somatic genetic variants [9,117].

## 6. Clinical and Laboratory Characteristics of MPNs Related to the 46/1 Haplotype

The relationship between JAK2V617F and the 46/1 haplotype is clear and has been pointed out by most studies, especially with patients with PV. This issue was confirmed by Ohyashiki [89], who identified a greater presence of haplotype variants, significantly elevated hemoglobin levels in patients with JAK2V617F and the GCC genotype compared to those with the GCC genotype, but without JAK2V617F. It is also interesting to note that some studies associate the haplotype with certain clinical findings (splenomegaly, splanchnic vein thrombosis, and Budd–Chiari syndrome) and laboratory findings (increased platelet, leukocyte, hematocrit, and hemoglobin counts) that are characteristic of MPNs [86,90,91,99,123,124]; however, this correlation is not a consensus [96,125]. Even so, it cannot be ignored that hemoglobin and hematocrit can be altered to levels above normal in cases of homozygosity and the high load of variant allele for JAK2V617F [125]; and the latter has already been shown to be related to 46/1 in several studies [86,125].

As discussed earlier in this review, homozygosity for the somatic variant is related to aUPD which, in turn, can be caused by 46/1 due to the combination of large portions of the two parental regions of chromosome 9p [96], i.e., these three elements (aUPD + HAPLOTYPE 46/1 + JAK2V617F), in theory, work together to establish the myeloproliferative and hereditary phenotype of MPNs. In the case of the latter, more specifically on the heritability of variants, it is believed that a marker allele situated on the same haplotype as a causative allele (JAK2V617F) will likely be inherited together, which would not be possible if the alleles are in different haplotypes [126].

## 7. Inheritance of MPNs and the Relationship with the 46/1 Haplotype

The development of a hematological neoplasm is dependent on several factors, such as age, environment, and host genetics [46]. Inherited genetic factors alter risk at each stage of development, i.e., from cancer acquisition to its progression [46,127]. The hereditary forms of MPNs can be divided into the following two main categories: (1) hereditary syndromes that affect a single lineage with Mendelian inheritance, high penetrance, and polyclonal hematopoiesis; and (2) hereditary predisposition to true MPNs, which are characterized by low penetrance, clonal hematopoiesis, presence of somatic variants (e.g., JAK2V617F), and risk of progression to acute myeloid leukemia (AML) [46,128].

In hereditary predisposition, there is a possibility of occurrence in two or more members of the same family and, in this context, the term “familial” is used to describe an unknown alteration of the germline that predisposes the acquisition of an MPN [46]. A family study of more than 11,000 patients with MPNs and their nearly 25,000 first-degree relatives found a 5–7-fold increased risk of developing MPNs among first-degree relatives of patients with MPNs [70]. Another survey, carried out with 72 families, characterized 50% of the individuals included in the study with an inheritance pattern consistent with autosomal dominance with incomplete penetrance [54].

There is growing evidence to suggest that hereditary factors are responsible for a broader effect on susceptibility to the development of MPNs [47,129]. Somatic variants seen in familial MPNs are responsible for the proliferative advantage and subsequent cellular clonality, while the inherited component predisposes one to the acquisition of somatic gene variations [46,130]. However, there are studies that do not agree with this direct relationship, and point out that other factors, in addition to the haplotype, would explain the inheritance of MPNs [44,46,109,115], such as, for example, the interaction with the rs2736100 (C) allele of the* TERT*, which is significantly found in familial MPNs when compared to sporadic MPNs [131]. Even if this portion disagrees with the propositions made, evaluating the variants that are part of the 46/1 germline haplotype in familial cases becomes relevant in order to understand its behavior within the heredity scenario and its possible relationship with the familial MPNs.

## 8. Conclusions and Perspectives

It is notable that the genetic scenario of MPNs is complex and still under elucidation. The 46/1 haplotype is an important finding in this discovery process, mainly due to its relationship with the JAK2V617F variant, and insertions and deletions of exon 12 of *JAK2* [88]. There is also a possible association with exon 10 variants of the *MPL *gene [132], and this is still under discussion [133,134], as well as the *CALR *gene [135,136]. Considering the haplotype as the object of analysis, the complexity of studying the region in which it is located cannot be excluded because, in addition to being considerably extensive, it has hundreds of SNVs in LD that are located in the intron regions, and which have been still scarcely studied. However, it would be naive to infer that within this region only one variant would be germline and responsible for the positivity of JAK2V617F or heredity of MPNs.

Interestingly, some of the associations with 46/1 described so far involve pathologies that are accentuated or characterized by inflammatory dysregulation: MPNs, AML, Chron’s disease, inflammatory bowel disease, and ulcerative colitis. These relationships are probably not random and further support the hypothesis that the 46/1 haplotype may be associated with JAK2V617F and/or other functional variants of the* JAK2* gene that have not yet been described and that play a role in inflammatory dysregulation [117].

The 46/1 haplotype may even establish itself as a viable alternative for monitoring individuals with MPNs and other myeloid neoplasms. Such an indication is considered for its association with shortened survival in patients with PMF due to reduced defense against infections and increased risk of a more severe inflammatory response, which, in turn, contribute to tissue remodeling in the bone marrow, thus, leading to a myelofibrotic transformation [24,56,58,97,137]; with the high variant allele load related to a more severe MPN phenotype; increased risk of myelofibrotic transformation in patients with PV [69]; and for being a possible factor related to AML with normal karyotype associated with predisposition to an acute myelomonocytic form, which makes it an unfavorable independent risk factor [117].

Another reported association would be with acute graft versus host disease (aGvHD) grades II-IV in AML patients undergoing allogeneic hematopoietic stem cell transplantation (allo-HSCT) [138]. Both this and the other studies cited must have their data confirmed by other studies with larger and more heterogeneous populations in order to verify the relationship between these germline variants and MPNs and other myeloid neoplasms. It is through these confirmations that such variants may become useful in clinical practice in order to achieve satisfactory results in therapy, as is already the case of haplotypes related to human leukocyte antigen (HLA) [139].

The use of 46/1 in screening or predictive tests in cases of familial MPNs may be an alternative to be considered. Even if there is no consensus on the impact of the haplotype in familial cases, its existence and relationship cannot be disregarded. The investigation of these variants becomes relevant due to the findings described so far and the association between early age at diagnosis in familial MPNs [140] and cases of childhood ET already reported in the literature [141]. The findings from these analyses can be useful for verifying the individual’s probability of carrying an inherited trait, which may or may not be pathogenic [142], in order to trace the molecular profile of childhood cases (genetic composition in childhood ET can be more complex than in adults) [141]. In addition, it makes it possible to predict disease susceptibility, favoring early diagnosis for preventive strategies and personalized therapies [143,144]. The analysis of haplotype variants in triple negative patients (who do not have driver variants in the *MPL*, *JAK2, *and* CARL* genes) can also be useful in this investigative context of MPNs. As established by the WHO, these patients should be tested for other variants in additional genes, such as *ASXL1, DNMT3A, TET2, EZH2, IDH1/2, *and *SRSF2,* in order to verify the nature of the clonal myeloproliferative disorder [52]. The inclusion of the haplotype in this screening analysis would help to provide a better understanding of the genetic scenario of these patients in order to verify whether the germline variants are involved in these mechanisms and interact with other genetic variants not yet known, in addition to proving or disproving their action within the context of dysregulation of inflammation in these cases. The use of next-generation sequencing (NGS) would be a valuable alternative in this process, and could help to confirm this diagnosis [145]. It would also allow the simultaneous assessment of the molecular complexity of the disease with greater coverage and sensitivity, as well as lower costs [52].

Diagnostics and prognostics based on DNA analysis applied to the clinical dynamics of cancer patients are constantly expanding and help us to understand the complexity of cancer genomes [85,88,146,147]. Although, currently, we have limited knowledge about the haplotype-phenotype interactions of genes involved in MPNs. The continuity of research on this topic, in different populations around the world and with different clinical and laboratory associations, will aid in a better understanding of the real impact of 46/1 on myeloproliferative dynamics. The results from these studies can produce useful tools in the diagnosis, personalized follow-up, genetic counseling, and training of the physician for decision-making about the planning and choice of treatment for the patient, thus, improving not only survival, but also the quality of life of the patient.

## Figures and Tables

**Figure 1 ijms-23-12582-f001:**
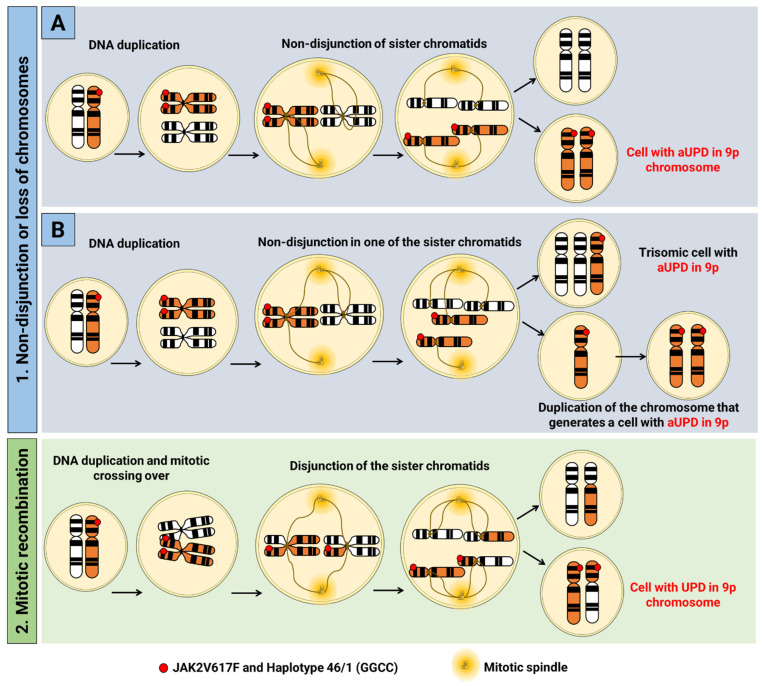
Mechanisms related to acquired uniparental disomy in hematopoietic pluripotent stem cells. This process can occur due to (1) (**A**) nondisjunction of sister chromatids or (**B**) delay in anaphase causing trisomy and monosomy of one of the chromosomes of the set, where in the cell with monosomy there is duplication of the remaining chromosome in the attempt to balance the loss of a chromosome, which results in two identical chromosomes in the same cell; or by (2) reciprocal exchange of chromosomal material during mitosis (mitotic recombination), such as chromatids, which generates several possible outcomes. In this example, applied to chromosome 9, the presence of the 46/1 haplotype and the JAK2V617F variant is illustrated, and is conditioned to the state of homozygosity.

**Figure 2 ijms-23-12582-f002:**
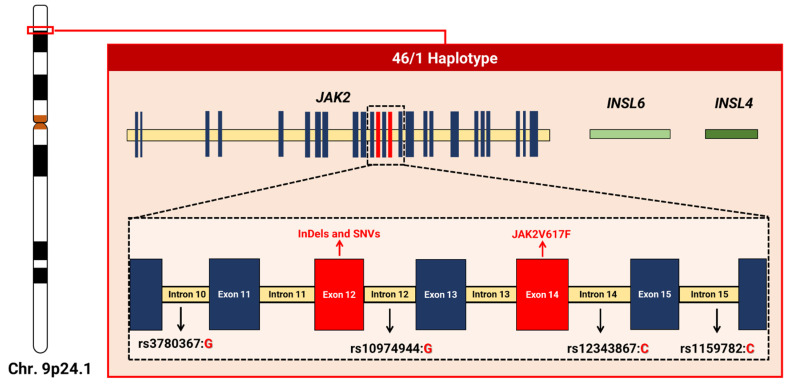
Schematic representation of the 46/1 haplotype based on the *JAK2* gene reference sequence (NG_009904). The haplotype extends over a block with approximately 250–280 kb on chromosome 9p, and encompasses three genes (*JAK2*, *INSL6*, and *INSL4*) and regions with a high rate of genetic variants in JAK2, such as exon 12 (with alterations such as insertions, deletions, and substitutions) and exon 14 (location of JAK2V617F). Four variants (rs3780367, rs10974944, rs12343867, and rs1159782) mark the haplotype and establish another nomenclature based on the variant alleles, GGCC, as also mentioned in the literature [97].

**Figure 3 ijms-23-12582-f003:**
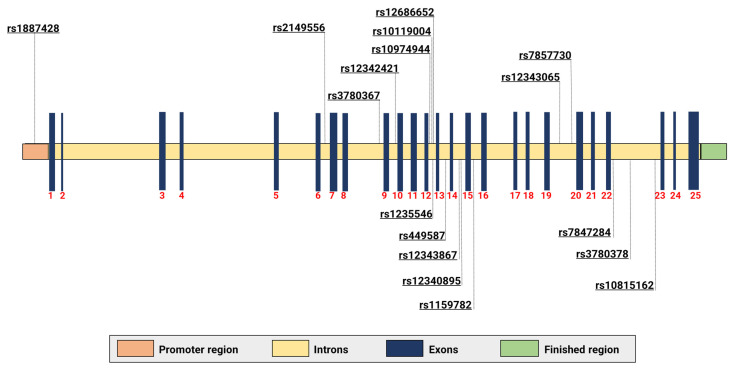
Location of variants identified in studies targeting the 46/1 haplotype. The mapping of variants along the gene was performed based on the reference sequence (NG_009904).

**Figure 4 ijms-23-12582-f004:**
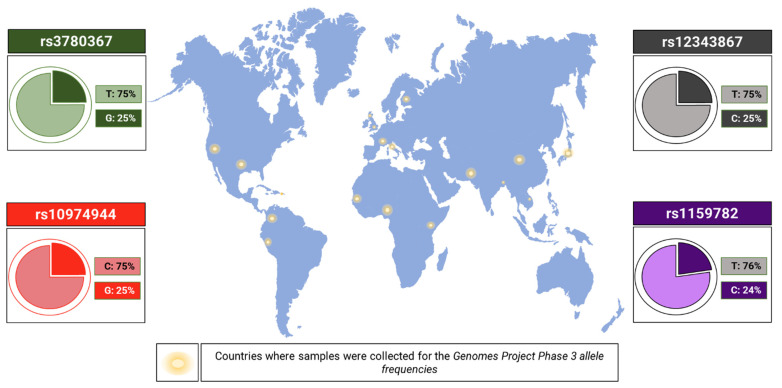
Minor allele frequency (MAF) of the 46/1 haplotype markers. Frequencies were measured from samples collected from individuals from Puerto Rico, Colombia, Peru, Gambia, Nigeria, Kenya, Italy, France, United Kingdom, Finland, Pakistan, India, China, Japan, and of Mexican origin residing in California and Texas (United States) [104,105,106,107,108].

**Figure 5 ijms-23-12582-f005:**
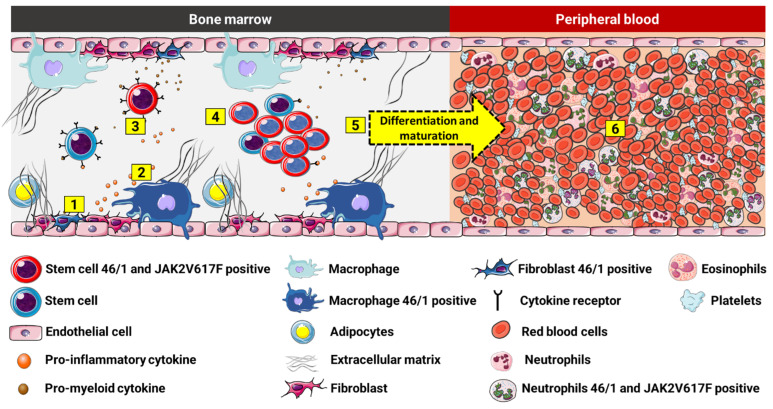
Possible association of the 46/1 haplotype and MPNs. (1) Medullary stromal cells positive for 46/1 may show dysregulation in genes, such as *INSL4* and *INSL6*, which may be involved in the (2) excessive production of proinflammatory and promyeloid mediators. These cytokines (3) interact with normal and haplotype-positive and JAK2V617F multipotent stem cells, promoting (4) exacerbated proliferation (proliferative advantage) of the mutated cells, which, in turn, continue their process of (5) differentiation and cell maturation, and trigger the (6) clonal myeloproliferative disorder.

**Table 1 ijms-23-12582-t001:** Features of *BCR-ABL1*-negative myeloproliferative neoplasms. MPNs: myeloproliferative neoplasms.

MPN	Clinical Description	Epidemiology
Polycythemia vera (PV)	Unregulated proliferation of erythroid series elements and increased granulocyte and thrombocyte counts (panmyelosis) [4,5,7]	Incidence of 0.5–4.0 cases per 100,000 Australian individuals [10], Europeans [11,12], Koreans [13,14], New Zealanders [15], and North Americans [11] aged between 60 and 70 years [5,14,15,16,17,18]
Essential thrombocythemia (ET)	Elevated number of platelets in peripheral blood (>450 × 10^9^/L), caused by megakaryocytic hyperplasia in the bone marrow, with alteration of other medullary sectors (erythrocytic or granulocytic) in a qualitative or quantitative way [4,5,19]	Affects individuals between the fifth and sixth decade of life with an incidence between 0.9–2.4 cases per 100,000 in North Americans [20], Koreans [13,14], and New Zealanders [14,15,20].
Primary myelofibrosis (PMF)	MPN with a worse prognosis, characterized by the proliferation of predominantly abnormal megakaryocytes and granulocytes in the bone marrow, deposition of reticulin fibers, and extramedullary hematopoiesis [4,5,21]	Affects individuals between the sixth and seventh decade of life [5] and has an incidence of 0.33 cases per 100,000 individuals per year in North America [15]; 0.4 cases per 100,000 in the Republic of Korea [14]; and 0.88 cases per 100,000 individuals in New Zealand [15].

## Data Availability

Not applicable.

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
