# Peer review of "The Contribution of *JAK2* 46/1 Haplotype in the Predisposition to Myeloproliferative Neoplasms"

_ijms, 2022, doi:10.3390/ijms232012582_

Round 1

Reviewer 1 Report

Dear Editor,

Thanks for inviting me to review the review article “JAK2 46/1 haplotype in the predisposition to chronic myeloproliferative neoplasms: We need to keep talking about it!”. In this article, authors discussed the main findings and discussions involving the 46/1 haplotype, its role in association with JAK2V617F, and highlighted the molecular and immunological aspects and their relevance as a tool for clinical practice and investigation of familial cases.

In general, this manuscript provided an overview of the relationship between JAK2 and 46/1 haplotype, and the cause of the genesis of chronic myeloproliferative neoplasms, however I think there are some ambiguous parts/areas that need to be answered/modified by authors.

1.     Regarding the title of the article: “JAK2 46/1 haplotype in the predisposition to chronic myeloproliferative neoplasms”, based on my understanding after reading the article, the JAK2 and 46/1 haplotype are two different causes of the chronic myeloproliferative neoplasms (MPNs), they  just have some associations with each other, but the title gives me an impression that the JAK2 and 46/1 haplotype are the same thing.

2.      Second, what is the purpose of discuss the Acquired uniparental dysomy? This part doesn’t seem highly relevant to the rest of the article to me, does this part is essential for the whole article?

Thanks!

Author Response

Dear Dr. Reviewer, 

First, I would like to thank you for your contributions to the improvement of the manuscript. I send the answers to the questions raised

1) Regarding the title of the article: “JAK2 46/1 haplotype in the predisposition to chronic myeloproliferative neoplasms”, based on my understanding after reading the article, the JAK2 and 46/1 haplotype are two different causes of the chronic myeloproliferative neoplasms (MPNs), they just have some associations with each other, but the title gives me an impression that the JAK2 and 46/1 haplotype are the same thing:

  • Title has been adjusted, as suggested. I would like to confirm that the 46/1 haplotype is within the JAK2 gene.

2. Second, what is the purpose of discuss the Acquired uniparental dysomy? This part doesn’t seem highly relevant to the rest of the article to me, does this part is essential for the whole article?

  • The discussion on uniparental disomy is essentially relevant for a better understanding of the article. Through this complex molecular process, genetic variants of the JAK2 gene can be conditioned to the homozygous state. Haplotype variants and also the JAK2V617F variant in the homozygous state are related to a more predominant symptomatology and greater severity of PWN. The topic complements information discussed in the following sections.

Reviewer 2 Report

Please, find attached

Author Response

Dear Dr. Reviewer, 

First, I would like to thank you for your contributions to the improvement of the manuscript. I send the answers to the questions raised

  • The title has been adjusted, as suggested by the reviewer;
  • All revisions and corrections were considered for the correction of the work, which were incorporated into the manuscript.
  • Regarding the phrases for reformulation, the change was made and can be seen in the body of the manuscript;
  •  The suggested revision on the names of genes and proteins was also carried out;
  • The term “DL” was used incorrectly and replaced by “LD”, which means linkage disequilibrium.